# Q2D2: A GEOMETRY-AWARE AUDIO CODEC LEVERAGING TWO-DIMENSIONAL QUANTIZATION

## ABSTRACT

Recent neural audio codecs have achieved impressive reconstruction quality, typically relying on quantization methods such as Residual Vector Quantization (RVQ), Vector Quantization (VQ) and Finite Scalar Quantization (FSQ). However, these quantization techniques limit the geometric structure of the latent space, make it harder to capture correlations between features leading to inefficiency in representation learning, codebook utilization and token rate. In this paper we introduce Two-Dimensional Quantization (Q2D2), a quantization scheme in which feature pairs are projected onto structured 2D grids—such as hexagonal, rhombic, or rectangular tiling—and quantized to the nearest grid values, yielding an implicit codebook defined by the product of grid levels, with codebook sizes comparable to conventional methods. Despite its simple geometric formulation, Q2D2 improves audio compression efficiency, with low token rates and high codebook utilization while maintaining state-of-the-art (SOTA) reconstruction quality. Specifically, Q2D2 achieves competitive to superior performance in various objective and subjective reconstruction metrics, across extensive experiments in speech domain compared to SOTA models. Comprehensive ablation studies further confirm the effectiveness of our design choices.

## 1 INTRODUCTION

In recent years, Large Language Models (LLMs) (Brown et al., 2020) have demonstrated remarkable progress in audio generation tasks, ranging from multi-speaker speech synthesis (Wang et al., 2023; Kharitonov et al., 2023; Jiang et al., 2023; Ji et al., 2024a) to music generation (Agostinelli et al., 2023) and general-purpose audio synthesis (Kreuk et al., 2022). At the same time, growing attention has been devoted to incorporating speech as a modality within large multimodal systems, as seen in models such as SpeechGPT (Zhang et al., 2023a), AnyGPT (Zhan et al., 2024), GPT-4o, GPT-5, and Moshi (Défossez et al., 2024). A key enabler of these advances has been the use of discrete acoustic representations produced by neural codecs (Zeghidour et al., 2021; Défossez et al., 2022; Kumar et al., 2023; Ji et al., 2024b). By converting high-rate speech signals into compact sequences of discrete tokens, acoustic codec models provide the crucial link between continuous audio and token-based language models, thereby enabling the direct application of LLM architectures to audio.

Most end-to-end discrete codec models (Défossez et al., 2022; Wu et al., 2023) adopt a three-stage structure consisting of an encoder, a RVQ module (Lee et al., 2022), and a decoder. The encoder performs downsampling of the audio signal in the time domain to obtain compressed audio frames. Each compressed audio frame is then quantized by a series of quantizers, with each quantizer operating on the residual of the previous one. The number of quantizers determines the overall bitrate. The decoder, on the other hand, performs upsampling in the time domain to reconstruct the audio signal from the quantizer outputs. Existing acoustic codec models (Kumar et al., 2023; Défossez et al., 2022; Siuzdak, 2023) demonstrate impressive reconstruction quality, and generative models based on discrete codecs are now capable of synthesizing speech at near-human levels. In response (Ji et al., 2024b) proposed a much simpler design: instead of stacked RVQ, it uses a single VQ layer (Gray, 1984) over features, showing that efficient tokenization can be achieved without deep quantizer hierarchies. Additional models have contributed to the expansion of the codec landscape. Some models (Pan et al., 2024; Yang et al., 2023; Zhang et al., 2023b) enhanced robustness, controllability, and synthesis quality through architectural and training innovations, while other models

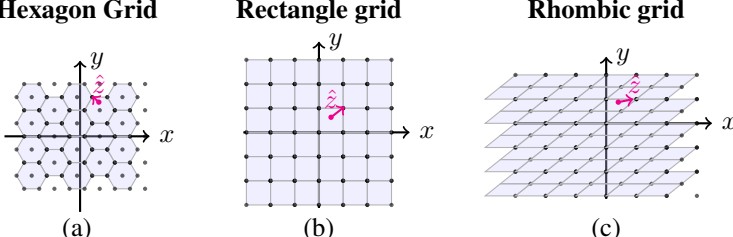

Figure 1: Visualization of quantization grids used in Q2D2: **Hexagonal Grid (a):** a hexagonal tiling with 9 quantization levels in $x$ and $y$ axis. **Rectangle Grid (b):** a rectangle tiling with 7 quantization levels in $x$ and $y$ axis. **Rhombic Grid (c):** a rhombic tiling with 7 quantization levels in $x$ axis, and 6 levels yielding to 11 quantization levels in $y$ axis.

(Li et al., 2024; Liu et al., 2024; Xin et al., 2024) aimed for universality and scalability, either by unifying audio and speech tasks under a single tokenizer or by increasing codec capacity. Complementary efforts refined training strategies, with stronger discriminators advancing adversarial learning (Ahn et al., 2024b;a).

Despite these successes, existing quantization schemes based on Vector Quantized-Variational AutoEncoder (VQ-VAE) and RVQ are challenging to optimize, and leads to well-documented problem of underutilized codebooks (Łańcucki et al., 2020; Takida et al., 2022a; Dhariwal et al., 2020; Huh et al., 2023a) as the codebook size is increased, many codewords will be unused. Subsequent works aimed to improve this with various tricks such as reinitializing the entire codebook or some codewords (Dhariwal et al., 2020; Łańcucki et al., 2020), stochastic formulations (Takida et al., 2022a), etc. At the other extreme, FSQ architecture (Mentzer et al., 2024) offered a strikingly simple alternative: each latent channel is quantized independently onto a fixed set of scalar levels, forming an implicit codebook given by the product of these sets. FSQ avoids collapse by design and guarantees high codebook utilization. However, because it quantizes each feature dimension separately, it results one-dimensional isolated quantization per channel, thereby being less effective in capturing correlations between feature dimensions.

**Our motivation is to retain the simplicity and utilization benefits of FSQ, while enriching the representational capacity of discrete audio codes.** In particular, we ask: can we capture correlations between channels without reintroducing the instability and inefficiency of high-dimensional vector quantization? Our approach is to move beyond simple 1D scalar grids by introducing structured 2D geometric tilings, and extending further to 3D geometric tilings in future work (Appendix E).

In this paper, we introduce **Q2D2**, a geometry-aware quantization scheme capable of reconstructing speech with low token rate. Instead of quantizing each latent channel independently, Q2D2 groups channels into pairs and maps them onto structured two-dimensional grids. Each pair is snapped to the nearest grid point from a fixed tiling (e.g., hexagonal, rhombic, or rectangle), producing an implicit codebook defined by the product of all pairwise grids. To interface with neural encoders, Q2D2 introduces lightweight linear projections into and out of the quantization space. Quantization itself is implemented with Straight-Through gradient Estimators (STE), and per-pair grid construction, ensuring differentiability, stability, and flexibility. This design achieve high utilization and robustness while introducing geometric structure that captures correlations between features and expands the expressive capacity of discrete codes. Our contributions can be summarized as follows:

1. **Conceptual Contributions.** We introduce **Q2D2** a novel approach of compressing the quantizer layers of acoustic codes models to a geometry-aware quantizer that groups channels into pairs and jointly quantizes them on utilizing complex geometric structures for the first time, enhancing semantic information of the codec and captures correlations between feature dimensions. Q2D2 supports per-pair quantization, levels selection, dimension selection, projection layers, and straight-through estimators, enabling efficient end-to-end training.

2. **Methodological Contributions.** We design a quantization space for compressing the codec model into a 2D single quantizer, testing multiple types of structured tilings for quantization, including hexagonal, rectangular, and rhombic grids, and demonstrating the impact of geometric shapes on representation quality and performance. Additionally, we examine

various quantization parameters such as resolution levels, projected dimension sizes and more, assessing their effects on performance and codebook utilization.

3. **Experimental Contributions.**: Q2D2 achieves competitive and surpasses speech reconstruction performance of SOTA models. It achieves comparable results with a very low tokens rate across broader metrics. Additional experiments demonstrate the high performance of Q2D2 over competitive baseline models regarding semantic information. Various ablation studies including grid design, dimension size and bandwidth, and level selection show that Q2D2 attains high codebook utilization without relying on auxiliary tricks such as commitment losses or codebook reseeding. Together, these results highlight Q2D2 as a simple yet powerful quantization method that unlocks richer discrete audio representations.

## 2 RELATED WORK

**Quantization methods.** The original VQ-VAE formulation (van den Oord et al., 2017) introduced a commitment loss together with Expectation-Maximization Attention (EMA) for stabilizing codebook learning. Later, (Roy et al., 2018) applied a soft Expectation Maximization (EM) approach, highlighting the role of codebook size tuning for different downstream tasks. VQ-VAE variants were quickly adopted in audio: (Dhariwal et al., 2020) used VQ-VAE for music generation, adding "random restarts" to prevent collapse and proposing a multi-scale hierarchy. Further improvements include periodically reinitializing codebooks via offline clustering (Łańcutki et al., 2020) and stochastic quantization schemes (Takida et al., 2022b; Williams et al., 2020), where noise or hierarchical structures are used to improve robustness. More recently, (Huh et al., 2023b) revisited training instabilities in VQ, proposing re-parameterization, alternating optimization, and a refined commitment loss. In addition to VQ-VAE, RVQ has proven effective in both image (Lee et al., 2022) and audio domains (Zeghidour et al., 2021), where residuals are recursively encoded by successive codebooks. Product Quantization (PQ) decomposes the latent space into subspaces with smaller codebooks (Chen et al., 2020; El-Nouby et al., 2022), while other work reduces token counts to improve inference efficiency (Huang et al., 2023). A distinct line of research introduces FSQ (Mentzer et al., 2024), which quantizes each latent channel independently onto a fixed scalar grid, forming an implicit product codebook. FSQ guarantees high utilization and avoids collapse entirely, though its strictly one-dimensional nature ignores inter-channel correlations.

Table 1: Comparison of VQ, FSQ, RVQ, and our Q2D2 quantization methods.

| Feature | VQ | RVQ | FSQ | Q2D2 |
|---|---|---|---|---|
| Quantization | $\arg\min_{c \in \mathbb{C}} \|\boldsymbol{z} - c\|$ | Sequential $\arg\min_{c \in \mathbb{C}_j} \|\boldsymbol{r}_{j-1} - c\|$ | $\text{round}(f(\boldsymbol{z}))$ | $\arg\min_{g \in \mathbb{G}_i} \|\boldsymbol{z}_{(i)} - g\|$ |
| Gradients | STE | STE | STE | STE |
| Auxiliary losses | Commitment, entropy | Commitment, entropy | – | – |
| Stabilization techniques | EMA, codebook splitting | EMA, codebook splitting | – | Projections |
| Codebook type | Explicit codebook $\mathbb{C}$ | Multiple explicit codebooks $\{\mathbb{C}_j\}$ | Implicit | Implicit |

**Neural audio codecs.** Recent codec models (Zeghidour et al., 2021; Défossez et al., 2022; Kumar et al., 2023; Ji et al., 2024b) have demonstrated the ability to reconstruct high-quality audio at low bitrates. These typically consist of an encoder that compresses the signal into latent features, a quantization stage, and a decoder that reconstructs the waveform. Acoustic tokens, unlike higher-level semantic tokens, preserve rich detail and generalize well across speech, audio, and music. This makes them particularly valuable for downstream generative models (Kharitonov et al., 2023; Huang et al., 2024a) and multimodal LLMs (Tongyi SpeechTeam, 2024; Anastassiou et al., 2024).

Within this family of approaches, several directions can be distinguished. Efforts to improve reconstruction quality include AudioDec (Wu et al., 2023), which highlighted the role of discriminators, PromptCodec (Pan et al., 2024), which enriched representations via auxiliary prompts, DAC (Kumar et al., 2023), which boosted fidelity with quantizer dropout and Short-Time Fourier Transform (STFT) based discriminators. Vocos (Siuzdak, 2023), which reduced artifacts through a pre-trained Encodec with an inverse Fourier vocoder, HILCodec (Ahn et al., 2024b), which proposed a new Multi-Filter Bank Discriminator (MFBD) to guide codec modeling, and APCodec (Ahn et al., 2024a), which incorporated ConvNextV2 modules for more powerful encoder–decoder modeling. Another line focuses on compression: HiFi-Codec (Yang et al., 2023) introduced parallel Group-Residual Vector Quantization (GRVQ) and achieved strong results with just four quantizers, while Language-Codec (Ji et al., 2024a) distributed information more evenly across quantizers us-

ing Masked Channel Residual Vector Quantization (MCRVQ), while Single-Codec (Li et al., 2024) demonstrated that competitive performance is possible even with a single quantizer.

Finally, some works aim to deepen understanding of the codec space. TiCodec (Ren et al., 2024) disentangled time-independent from time-dependent information, FACodec (Ju et al., 2024) decomposed codec latents into content, style, and acoustic modules, and several recent models explicitly integrate semantic representations. RepCodec (Huang et al., 2024b) learns a vector quantization codebook by reconstructing speech representations from speech encoders like HuBERT (Hsu et al., 2021) and Data2Vec (Baevski et al., 2022). SpeechTokenizer (Zhang et al., 2023b) enriched quantizer semantics through distillation, FunCodec (Du et al., 2023) made semantic tokens optional, and SemanticCodec (Liu et al., 2024) reconstructed audio from semantic tokens using a diffusion-based decoder. While these methods add semantic richness, they move away from the classical encoder–quantizer–decoder paradigm and introduce extra complexity.

**Comparison.** Relative to these approaches, Q2D2 achieves strong reconstruction with only a single quantizer and through compact token sequences (53, 166, or 333 tokens per second). By contrast, DAC (Kumar et al., 2023) requires about 900 tokens per second, spread across 9 quantizers.

## 3 METHOD

Our proposed **two-dimensional quantization (Q2D2)**, built on the framework of WavTokenizer (Ji et al., 2024b) (as described in A, groups latent feature channels into pairs and jointly quantizes them on structured two-dimensional grids such as hexagonal, rhombic, or rectangular tilings. As in other quantization schemes, the encoder and decoder absorb much of the non-linearity, but Q2D2 provides a richer geometric structure in the discrete space. For comparison, VQ learns a Voronoi partition of the high-dimensional latent space of VQ-VAE, which produces highly complex non-linear partitioning of the VQ-VAE (e.g. audio). In contrast, FSQ applies a simple fixed grid partition in much lower-dimensional space, but ignoring inter-channel structure. Q2D2 aim bridges these approaches by combining the robustness of FSQ with the expressive capacity of multi-dimensional grids. A side-by-side illustration of Q2D2, FSQ, and VQ is shown in Figure 2, and Table 1.

### 3.1 TWO-DIMENSIONAL QUANTIZATION

Let $\mathbf{x}$ denote the encoder's final-layer output. A learned affine projection maps $\mathbf{x}$ to $\mathbb{R}^d$, after which a hyperbolic tangent nonlinearity is applied, yielding $\mathbf{z} \in [-1, 1]^d$. Constraining the projected features to this interval facilitates subsequent alignment with the quantization grids, where $d$ is the chosen dimensional representation.

We require the dimensional representation $d$ to be even so that it can be reshaped into two-dimensional feature pairs $P = \frac{d}{2}$.

We first apply a bounding function so each channel is rescaled by a factor $l_i/2$, where $l_i \in \mathbb{N}$ is the number of quantization levels chosen for each dimension $i \in \{1, \ldots, d\}$. Formally,

$$z_i' = z_i \frac{l_i}{2}, \qquad i = 1, \ldots, d, \tag{1}$$

so that, the $z_i' \in \left[ -\frac{l_i}{2}, \frac{l_i}{2} \right]$ for each $i$. After the bounding step, Q2D2 reshapes $\mathbf{z}'$ into pairs of feature dimensions, where $z''_j$ is pair of features, and $1 \le j \le P$:

$$\mathbf{z}'' = \{ z_1'', \ldots, z_P'' \} = \{ (z_1', z_2'), \ldots, (z_{d-1}', z_d') \} \tag{2}$$

$z''_j$ are then jointly quantized to the nearest point in a fixed two-dimensional grid $\mathbb{G}_j$:

$$\hat{z}_j'' = \arg \min_{g \in \mathbb{G}_j} \|z''_j - g\|_2, \quad \mathbb{G}_j \subset \mathbb{R}^2, \tag{3}$$

Each grid $\mathbb{G}_j$ is instantiated according to a prescribed tiling scheme—hexagonal (Alg. 1), rectangular (Alg. 2), or rhombic (Alg. 3), the number of levels $l_i$, and the spread factor of the grid $e_i = \frac{l_i-1}{2}$. For grids visualization refer to figure 1.

The overall codebook is represented by the combination of the per-pair grids $\mathbb{G}_1, \ldots, \mathbb{G}_P$.

Where each pair has $L_j$ points:

$$L_j = l_{2j-1} \cdot l_{2j}, \qquad j = 1, \ldots, P \tag{4}$$

Yielding a total size:

$$|\mathbb{C}| = \prod_{j=1}^{P} L_j \tag{5}$$

Thus, Q2D2 defines an implicit structured codebook without the need to learn embeddings. To integrate with neural encoders, we use lightweight linear out projection of the quantization space. As in FSQ and VQ, gradients are propagated using STE. This design preserves high codebook utilization and robustness, while capturing correlations between features through structured 2D grids. Q2D2 quantization process is illustrated in Fig. 2 compared to FSQ and VQ.

Tiling algorithms: The following algorithms are pseudocode for construction of the **hexagonal (left)**, **rectangle (up right) and rhombic (down right)** tiling grids. In the pseudocode $l_{2j}$ and $l_{2j-1}$ denote the number of levels for the $x$ and $y$ axes per pair, respectively; $e_{2j-1}$ and $e_{2j}$ are their spread factors; $y_c$ and $x_c$ the coordinate grids; $x_o$ the offset of the grid ix $x$ axis; $g_h$ the hexagon tiling; $g_s$ the rectangle grid; $g_m$ the midpoints; $\mathrm{mg}\,(x,y)$ forms all coordinate pairs $(x, y)$ on a 2D lattice.

---

**Algorithm 1** Hexagonal grid

1: Input: $l_j, e_j$; Require: $l_j \geq 2$
2: $dx \leftarrow \frac{2e_j}{l_j-1}$; $\quad dy \leftarrow dx \cdot \frac{\sqrt{3}}{2}$
3: $y_c \leftarrow$ uniform grid in $[-e_j, e_j]$ of length $l_j$
4: **for** $i, y \in$ enumerate$(y_c)$ **do**
5: $\quad x_o \leftarrow \begin{cases} -dx/4 & i \bmod 2 = 1 \\ dx/4 & \text{else} \end{cases}$
6: $\quad x_c \leftarrow$ uniform grid in $[-e_j, e_j] + x_o$
7: $\quad$ append mg$(x_c, y)$ to $\mathbb{G}_j$
8: **end for**
9: output: concat$(\mathbb{G}_j)$

---

**Algorithm 2** Rectangle grid

1: Input: $l_{2j-1}, l_{2j}, e_{2j-1}, e_{2j}$; Require $l_{2j-1}, l_{2j} \geq 2$
2: $c_x \leftarrow$ uniform grid in $[-e_{2j-1}, e_{2j-1}]$
3: $c_y \leftarrow$ uniform grid in $[-e_{2j}, e_{2j}]$
4: $\mathbb{G}_j \leftarrow$ flatten(mg$(c_x, c_y)$)
5: output: $\mathbb{G}_j$

---

**Algorithm 3** Rhombic grid

1: Input: $l_{2j-1}, l_{2j}, e_{2j-1}, e_{2j}$; Require $l_{2j-1}, l_{2j} \geq 2$
2: $dx \leftarrow \frac{2e_{2j-1}}{l_{2j-1}-1}$; $dy \leftarrow \frac{2e_{2j}}{l_{2j}-1}$
3: $c_x \leftarrow$ uniform grid in $[-e_{2j-1}, e_{2j-1}]$
4: $c_y \leftarrow$ uniform grid in $[-e_{2j}, e_{2j}]$
5: $g_s \leftarrow$ flatten(mg$(c_x, c_y)$)
6: $g_m \leftarrow$ flatten(mg$(c_x + dx/2, c_y + dy/2)$)
7: $\mathbb{G}_j \leftarrow$ concat$(g_s, g_m)$
8: output: $\mathbb{G}_j$

---

## 3.2 HYPERPARAMETERS

Q2D2 is governed by three sets of hyperparameters: (i) the dimension of the feature $d$ (which must be even), (ii) the geometry of the grid (e.g., rectangle, hexagonal, rhombic) and (iii) the number of levels per pair of features $L_j = [L_1, \ldots, L_P]$. The effective codebook size is the product $\prod_j L_j$, comparable to VQ or FSQ codebooks of similar scale.

## 3.3 PARAMETER COUNT

Like FSQ, Q2D2 avoids a learned codebook. The main savings over VQ come from not learning a codebook of size $|C| \cdot d$ (e.g., $|C| = 2^{12} = 4096$ and $d = 512$ implies $\sim 2$M parameters) and from using a smaller latent dimension, which also reduces encoder size. Q2D2 uses fixed analytic grids, so the effective codebook size $\prod_j L_j$ does not add parameters. The only learnables are lightweight projection layers scaling with $d$, not with $|C|$ or $\prod_j L_j$. Thus, for the same codebook size, both Q2D2 and FSQ yield smaller models than VQ, with parameter count dominated by $d$.

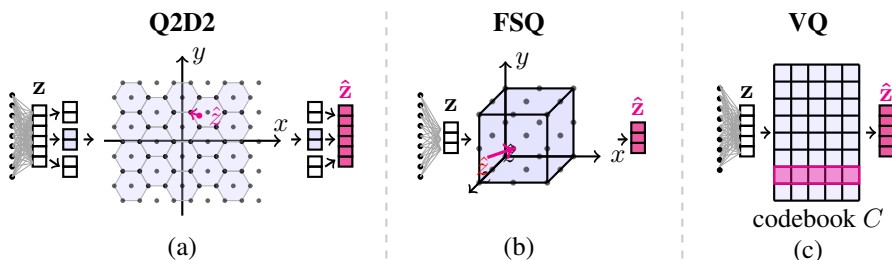

(a)        (b)        (c)

Figure 2: **Q2D2 (a):** The final encoder layer is projected to $d$ selected latent feature dimensions. Each projected dimension is first bounded between $[-l_i/2, l_i/2]$, where $l_i$ is the number of levels selected per dimension. Q2D2 then groups the dimensions into pairs (in example, 6 dimensions are reshaped into 3 pairs), and jointly quantizes each pair onto a structured 2D grid and finding the nearest point on the grid. **FSQ (b):** The final encoder layer is projected to $d$ dimensions (example with $d = 3$). Each projected dimension $z$ is bounded to $l$ discrete values (here $l = 3$), and then rounded to the nearest integer, producing the quantized vector $\hat{z}$, the nearest point in the hypercube. **VQ (c):** The final encoder layer is projected to $d$ dimensions (example shown with $d = 5$, as $d$ is typically larger in VQ). The latent vector $z$ is replaced by the closest vector from the codebook $\hat{z} \in \mathbb{C}$ via nearest-neighbor lookup.

## 4 EXPERIMENTS

### 4.1 EXPERIMENTAL SETUP

**Datasets.** The training for the main experiments in Table 2 was conducted on approximately 8K hours of data, following the setup used in WavTokenizer (Ji et al., 2024b). For the speech domain, we use LibriTTS (Zen et al., 2019), VCTK (Veaux et al., 2016), and a randomly selected 3000-hour subset of CommonVoice (Ardila et al., 2019). For the general audio domain, we utilize a 2000-hour subset of AudioSet (Gemmeke et al., 2017), and for the music domain, we employ the Jamendo (Bogdanov et al., 2019) and MUSDB18 (Rafii et al., 2017) datasets.

For evaluation between Q2D2 and other baselines (Table 3) we train the models on approximately 150k hours of multilingual speech, drawn from the Emilia dataset (En/Zh/De/Fr/Ja/Ko) (He et al., 2024) and MLS (En/Fr/De/Nl/Es/It/Pt/Pl) (Pratap et al., 2020).

For the ablation studies, and comparison between Q2D2 to FSQ and WavTokenizer (vQ) (Table 4), we train all models on the LibriTTS corpus (Zen et al., 2019), which contains approximately 585 hours of English speech. Speech reconstruction performance is evaluated under both clean and noisy conditions using the *test-clean* and *test-other* subsets, respectively.

**Baselines.** We evaluate Q2D2 against large set of neural audio codecs: WavTokenizer (Ji et al., 2024b) was selected as a primary baseline due to its SOTA performance at low bitrates with single quantization layer, Encodec (Défossez et al., 2022), Vocos (Siuzdak, 2023), SpeechTokenizer (Zhang et al., 2023b), DAC (Kumar et al., 2023) and HiFi-Codec (Yang et al., 2023). To compare with WavTokenizer (Ji et al., 2024b) and SOTA baselines compared in WavTokenizer paper, we trained our model on the 8K hours WavTokenizer dataset.

We also evaluate our models against X-codec2 (Ye et al., 2025), DAC (Ye et al., 2024), WavTokenizer (Ji et al., 2024b), Encodec (Défossez et al., 2022), SpeechTokenizer (Zhang et al., 2023b), DAC (Kumar et al., 2023), Mimi (Défossez et al., 2024), StableCodec (Parker et al., 2024), SemantiCodec (Liu et al., 2024), trained on 150k hours of multilingual speech, drawn from the Emilia dataset (En/Zh/De/Fr/Ja/Ko) (He et al., 2024) and MLS (En/Fr/De/Nl/Es/It/Pt/Pl) (Pratap et al., 2020).

**Evaluation Metrics.** For objective evaluation of discrete codec models, following WavToknizer (Ji et al., 2024b), we employ UTMOS (Saeki et al., 2022) automatic Mean Opinion Score (MOS) prediction system. UTMOS can yield scores highly correlated with human evaluations, closer to human perception than PESQ (Rix et al., 2001) Perceptual Evaluation of Speech Quality, but it is restricted to 16 kHz sample rate. We also adopt the metrics in speech enhancement fields, such as PESQ, STOI (Taal et al., 2011) Short-Time Objective Intelligibility, and the V/UV F1 score (Siuzdak et al., 2018) for voiced/unvoiced classification. In addition to these objective metrics, following Encodec (Défossez et al., 2022) and WavTokenizer (Ji et al., 2024b), we employ the

*subjective* MUSHRA evaluation to assess the reconstruction performance of the codec, and also common subjective Comparison Mean Opinion Score (CMOS) evaluation metrics. Details of the subjective evaluation protocols are provided in Appendix C.

**Implementation and Setup Details.** In our experiments, we found that $11 \geq l_i \geq 5$ quantization levels for all dimensions yields stable performance, while rhombic grids offer higher packing efficiency at a light more level count then hexagonal and rectangle, with $d = 6$ projection feature dimensions. Due computational resource constraints we train some of Q2D2 models on 2 NVIDIA RTX6000 48G GPUs and others on 2 NVIDIA L40S 48G GPUs approximately 40 epochs per model. Throughout the entire training process, all input speech and audio samples resampled to 24 kHz, with batch size equal to 16, and was optimized using the AdamW optimizer. During training we used initial learning rate of $8e^{-5}$ with decay based on a cosine schedule. More effect of different models and design choices were analyzed in Sec. 5.

## 4.2 MAIN RESULTS

**Evaluation on Reconstruction**. We compare speech reconstruction performance of Q2D2 (trained on 8K Wavtokenizer dataset) with large selection of SOTA competitive codec models WavTokenizer, Encodec, DAC, Vocos, SpeechTokenizer and HiFi-Codec (trained on WavTokenizer large dataset) as baselines on LibriTTS test-clean (4837 samples), LibriTTS test-other (5120 samples), and LJSpeech (13100 samples), which correspond to audio reconstruction in clean, noisy, and out-of-domain environments, respectively. The results are shown in Table 2. **We observe the following:** Q2D2 with a rhombic grid at 3.3 kbps achieves strong performance on the UTMOS metric, surpassing all current SOTA models in the 0.5–9 kbps range on both LibriTTS-test-clean and LibriTTS-test-other. Since UTMOS is highly correlated with human perception of audio quality (Saeki et al., 2022), this demonstrates that Q2D2 preserves perceptual quality even under low comparison. At 3 kbps, Q2D2 with only 166 tokens consistently outperforms competing models that rely on over 300 tokens and multiple quantizers, across UTMOS, PESQ, STOI, and F1 (with the exception of UTMOS on LJSpeech). Likewise, the 6.9 kbps Q2D2 model with a rhombic grid surpasses all SOTA models at 6 kbps, outperforming models with 600 tokens while using only 333 tokens, across all metrics and datasets. Finally, when compared to single-quantizers baselines, Q2D2 at 1 kbps, with only 75 tokens outperforms DAC (100 tokens) across all metrics on all test sets, and surpasses the WavTokenizer baseline (75 tokens) in PESQ, STOI, and F1 on LibriTTS-test-clean (illustrated in Figure 3) and LibriTTS-test-other (except PESQ on LibriTTS-test-other).

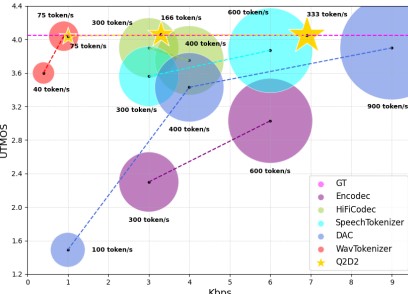

Figure 3: Comparison between different acoustic codec models. The y-axis UTMOS reflects reconstruction quality (UTMOS highly correlates with human evaluations), the x-axis kbps represents audio compression levels. The size of circles represents the number of discrete tokens per second.

We further evaluate reconstruction performance by comparing Q2D2 (trained on Emilia and MLS) against a suite of competitive SOTA codec models—Encodec, DAC, SpeechTokenizer, Mimi, X-Codec, BigCodec, WavTokenizer, Mini, StableCodec, SemantiCodec, and X-codec2—trained on the same data, using LibriSpeech test-clean (2620 samples) (Panayotov et al., 2015). The results are presented in Table 3. On this benchmark, Q2D2 continues to exhibit the same trends observed on LibriTTS and LJSpeech. With 333 tokens and a single quantizer, Q2D2 achieves the highest UTMOS and STOI scores among all models using more than 150 tokens. With 166 tokens, Q2D2 outperforms all models in the 100–150 token range in both PESQ and STOI, while maintaining competitive UTMOS performance. Finally, the 66-token Q2D2 configuration delivers strong overall performance across all metrics and achieves the highest STOI score within the ultra-low-token setting.

Table 2: Objective reconstruction results of various codec models on *LibriTTS test-clean* (clean environment), *LibriTTS test-other* (noisy environment), and *LJSpeech dataset* (out-of-domain environment). **Nq** denotes number of quantizers. **GT** denotes ground truth waveforms. Best results from models are in bold.

| Dataset | Model | Bandwidth↓ | Nq↓ | token/s↓ | UTMOS↑ | PESQ↑ | STOI↑ | V/UV F1↑ |
|---|---|---|---|---|---|---|---|---|
| | GT | - | - | - | 4.0562 | - | - | - |
| | DAC | 9.0kbps | 9 | 900 | 3.9097 | 3.9082 | 0.9699 | 0.9781 |
| | Encodec | 6.0kbps | 8 | 600 | 3.0399 | 2.7202 | 0.9391 | 0.9527 |
| | Vocos | 6.0kbps | 8 | 600 | 3.6954 | 2.8069 | 0.9426 | 0.9437 |
| | SpeechTokenizer | 6.0kbps | 8 | 600 | 3.8794 | 2.6121 | 0.9165 | 0.9495 |
| | Q2D2 | 6.9kbps | 1 | 333 | **4.0321** | **3.7006** | **0.9637** | **0.9799** |
| | DAC | 4.0kbps | 4 | 400 | 3.4329 | 2.7378 | 0.9280 | 0.9572 |
| | HiFi-Codec | 4.0kbps | 4 | 400 | 3.7529 | 2.9611 | 0.9405 | 0.9617 |
| LibriTTS test-clean | HiFi-Codec | 3.0kbps | 4 | 300 | 3.9035 | 3.0116 | 0.9446 | 0.9576 |
| | Encodec | 3.0kbps | 4 | 300 | 2.3070 | 2.0517 | 0.9007 | 0.9198 |
| | Vocos | 3.0kbps | 4 | 300 | 3.5390 | 2.4026 | 0.9231 | 0.9358 |
| | SpeechTokenizer | 3.0kbps | 4 | 300 | 3.5632 | 1.9311 | 0.8778 | 0.9273 |
| | Q2D2 | 3.3kbps | 1 | 166 | **4.0613** | **3.3635** | **0.9557** | **0.9676** |
| | DAC | 1.0kbps | 1 | 100 | 1.4940 | 1.2464 | 0.7706 | 0.7941 |
| | WavTokenizer | 0.5kbps | 1 | 40 | 3.6016 | 1.7027 | 0.8615 | 0.9173 |
| | WavTokenizer | 0.9kbps | 1 | 75 | 4.0486 | 2.3730 | 0.9139 | 0.9382 |
| | Q2D2 | 1kbps | 1 | 75 | **4.0526** | **2.5091** | **0.9217** | **0.9440** |
| | GT | - | - | - | 3.4831 | - | - | - |
| | DAC | 9.0kbps | 9 | 900 | 3.3566 | 3.7595 | 0.9576 | 0.9696 |
| | Encodec | 6.0kbps | 8 | 600 | 2.6568 | 2.6818 | 0.9241 | 0.9338 |
| | Vocos | 6.0kbps | 8 | 600 | 3.1956 | 2.5590 | 0.9209 | 0.9202 |
| | SpeechTokenizer | 6.0kbps | 8 | 600 | 3.2851 | 2.3269 | 0.8811 | 0.9205 |
| | Q2D2 | 6.9kbps | 1 | 333 | **3.4481** | **3.4595** | **0.9464** | **0.9712** |
| | DAC | 4.0kbps | 4 | 400 | 2.9448 | 2.5948 | 0.9083 | 0.9404 |
| | HiFi-Codec | 4.0kbps | 4 | 400 | 3.0750 | 2.5536 | 0.9126 | 0.9387 |
| LibriTTS test-other | HiFi-Codec | 3.0kbps | 4 | 300 | 3.3034 | 2.6083 | 0.9166 | 0.9318 |
| | Encodec | 3.0kbps | 4 | 300 | 2.0883 | 2.0520 | 0.8835 | 0.8926 |
| | Vocos | 3.0kbps | 4 | 300 | 3.0558 | 2.1933 | 0.8967 | 0.9051 |
| | SpeechTokenizer | 3.0kbps | 4 | 300 | 3.0183 | 1.7373 | 0.8371 | 0.8907 |
| | Q2D2 | 3.3kbps | 1 | 166 | **3.5072** | **3.0960** | **0.9339** | **0.9531** |
| | DAC | 1.0kbps | 1 | 100 | 1.4986 | 1.2454 | 0.7505 | 0.7775 |
| | WavTokenizer | 0.5kbps | 1 | 40 | 3.0545 | 1.6622 | 0.8336 | 0.8953 |
| | WavTokenizer | 0.9kbps | 1 | 75 | 3.4312 | **2.2614** | 0.8907 | 0.9172 |
| | Q2D2 | 1kbps | 1 | 75 | **3.5383** | 2.2224 | **0.8908** | **0.9199** |
| | GT | - | - | - | 4.3794 | - | - | - |
| | DAC | 9.0kbps | 9 | 900 | 4.3007 | 3.9022 | 0.9733 | 0.9757 |
| | Encodec | 6.0kbps | 8 | 600 | 3.2286 | 2.6633 | 0.9441 | 0.9555 |
| | Vocos | 6.0kbps | 8 | 600 | 4.0332 | 2.9258 | 0.9497 | 0.9459 |
| | SpeechTokenizer | 6.0kbps | 8 | 600 | 4.2373 | 2.6413 | 0.9316 | 0.9452 |
| | Q2D2 | 6.9kbps | 1 | 333 | **4.3302** | **3.4874** | **0.9658** | **0.9753** |
| | DAC | 4.0kbps | 4 | 400 | 3.8109 | 2.7616 | 0.9338 | 0.9524 |
| | HiFi-Codec | 4.0kbps | 4 | 400 | 4.1656 | 2.7629 | 0.9446 | 0.9497 |
| LJSpeech | HiFi-Codec | 3.0kbps | 4 | 300 | 4.2692 | 2.9091 | 0.9485 | 0.9469 |
| | Encodec | 3.0kbps | 4 | 300 | 2.3905 | 2.0194 | 0.9058 | 0.9326 |
| | Vocos | 3.0kbps | 4 | 300 | 3.7880 | 2.5006 | 0.9310 | 0.9388 |
| | SpeechTokenizer | 3.0kbps | 4 | 300 | 3.9908 | 2.0458 | 0.9021 | 0.9299 |
| | Q2D2 | 3.3kbps | 1 | 166 | **4.2909** | **3.2179** | **0.9552** | **0.9580** |
| | DAC | 1.0kbps | 1 | 100 | 1.4438 | 1.2084 | 0.7822 | 0.8095 |
| | WavTokenizer | 0.5kbps | 1 | 40 | 4.0186 | 2.1142 | 0.9093 | 0.9406 |
| | WavTokenizer | 0.9kbps | 1 | 75 | **4.2580** | **2.4923** | **0.9312** | **0.9397** |
| | Q2D2 | 1kbps | 1 | 75 | 3.9715 | 2.1914 | 0.9158 | 0.9231 |

To directly compare Q2D2 with FSQ and VQ (WavTokenizer), we evaluate speech reconstruction performance within the WavTokenizer framework, modifying only the quantization layer (Table 4). This setup enables a fair and controlled comparison across all three quantization methods. **Our results show that** Q2D2 at 1 kbps (75 tokens) consistently outperforms both FSQ and VQ across all metrics, demonstrating its stronger capability for high-quality reconstruction.

**Subjective Evaluation.** Following Encodec and WavTokenizer, we used MUSHRA (ITU-R, 2001) as the one of the metrics for subjective evaluation. As shown in Table 5, Q2D2 at 3.3 kbps outperforms the SOTA DAC model at 9 kbps in reconstruction quality on speech domain. Further more, Q2D2 at 1 kbps achieve a similar reconstruction quality as SOTA WavTokenizer at 0.9 kbps. As used in WavTokenizer, we utilized another subjective evaluation, focusing on two perceptual dimensions: audio Comparison Mean Opinion Score Quality (CMOS-Q) and Comparison Mean Opinion Score Prosody (CMOS-P). As shown in Table 6, Q2D2 achieves ratings closer to GT than WavTokenizer, with consistently higher CMOS-Q and CMOS-P scores. Both experiments result demonstrate that **Q2D2 is capable of maintaining high subjective reconstruction performance on speech domain with limited number of tokens.**

**Evaluation of Semantic Representation.** Following WavTokenizer steps, we evaluate the semantic richness of different codec models on the ARCH benchmark (La Quatra et al., 2024). The ARCH benchmark comprises 12 datasets in speech, music, audio domains (details in Appendix B). We extract embeddings corresponding to the discrete codebooks of an acoustic codec model as its respec-

Table 3: Objective reconstruction results of various codec models on *LibriSpeech test-clean* (clean environment). **Nq** denotes number of quantizers. **GT** denotes ground truth waveforms. Best results from models are in bold.

| Dataset | Model | Nq ↓ | token/s ↓ | UTMOS ↑ | PESQ ↑ | STOI ↑ |
|---|---|---|---|---|---|---|
| | GT | - | - | 4.09 | - | - |
| | DAC | 12 | 600 | 4.00 | **4.15** | 0.95 |
| | Encodec | 8 | 600 | 3.09 | 3.18 | 0.94 |
| | Q2D2 | 1 | 333 | **4.07** | 3.79 | **0.96** |
| | Encodec | 2 | 150 | 1.58 | 1.94 | 0.85 |
| | DAC | 2 | 100 | 1.29 | 1.40 | 0.73 |
| | SpeechTokenizer | 2 | 100 | 2.28 | 1.59 | 0.77 |
| | Mimi | 8 | 100 | 3.56 | 2.80 | 0.91 |
| | X-codec | 2 | 100 | **4.21** | 2.88 | 0.86 |
| | Q2D2 | 1 | 166 | 4.07 | **3.36** | **0.95** |
| LibriSpeech test-clean | BigCodec | 1 | 80 | **4.11** | **3.27** | **0.93** |
| | WavTokenizer | 1 | 75 | 3.79 | 2.63 | 0.90 |
| | Mimi | 6 | 75 | 3.38 | 2.51 | 0.89 |
| | Encodec | 1 | 75 | 1.25 | 1.48 | 0.77 |
| | DAC | 1 | 50 | 1.25 | 1.20 | 0.62 |
| | SpeechTokenizer | 1 | 50 | 1.27 | 1.30 | 0.64 |
| | Mimi | 4 | 50 | 3.03 | 2.09 | 0.85 |
| | StableCodec | 2 | 50 | **4.23** | 2.91 | 0.91 |
| | SemantiCodec | 2 | 50 | 2.71 | 2.18 | 0.84 |
| | X-codec | 1 | 50 | 4.05 | 2.38 | 0.83 |
| | X-codec2 | 1 | 50 | 4.13 | **3.04** | **0.92** |
| | WavTokenizer | 1 | 40 | 3.57 | 2.06 | 0.85 |
| | Q2D2 | 1 | 66 | 4.04 | 2.50 | 0.92 |

Table 4: Objective reconstruction results of Q2D2, WavTokenizer (VQ) and FSQ on *LibriTTS test-clean* (clean environment), *LibriTTS test-other* (noisy environment), and *LJSpeech dataset* (out-of-domain environment). **Nq** denotes number of quantizers. **GT** denotes ground truth waveforms. Best results from models are in bold.

| Dataset | Model | Bandwidth ↓ | Nq ↓ | token/s ↓ | UTMOS ↑ | PESQ ↑ | STOI ↑ | V/UV F1 ↑ |
|---|---|---|---|---|---|---|---|---|
| | GT | - | - | - | 4.0562 | - | - | - |
| | FSQ | 1kbps | 1 | 75 | 3.9929 | 2.3873 | 0.9163 | 0.9421 |
| LibriTTS test-clean | WavTokenizer | 0.5kbps | 1 | 40 | 3.5780 | 1.7088 | 0.8648 | 0.9172 |
| | WavTokenizer | 0.9kbps | 1 | 75 | 3.9665 | 2.4655 | 0.9188 | 0.9390 |
| | Q2D2 | 1kbps | 1 | 75 | **4.0483** | **2.5021** | **0.9218** | **0.9449** |
| | GT | - | - | - | 3.4831 | - | - | - |
| | FSQ | 1kbps | 1 | 75 | 3.4529 | 2.0974 | 0.8835 | 0.9157 |
| LibriTTS test-other | WavTokenizer | 0.5kbps | 1 | 40 | 3.0535 | 1.6622 | 0.8332 | 0.8949 |
| | WavTokenizer | 0.9kbps | 1 | 75 | 3.4302 | **2.2611** | 0.8904 | 0.9171 |
| | Q2D2 | 1kbps | 1 | 75 | **3.5303** | 2.2168 | **0.8909** | **0.9203** |
| | GT | - | - | - | 4.3794 | - | - | - |
| | FSQ | 1kbps | 1 | 75 | 3.7326 | 2.0568 | 0.9075 | 0.9191 |
| LJSpeech | WavTokenizer | 0.5kbps | 1 | 40 | 3.6838 | 1.6708 | 0.8706 | 0.9189 |
| | WavTokenizer | 0.9kbps | 1 | 75 | 3.8714 | 1.9516 | 0.8996 | 0.9101 |
| | Q2D2 | 1kbps | 1 | 75 | **3.9412** | **2.1749** | **0.9151** | **0.9227** |

tive representations and evaluate the classification accuracy of the codec model on ARCH datasets using its representations. We used the experimental results of WavTokenizer, Encodec and DAC from (Ji et al., 2024b) to compare to our results. Because our model trained only on LibriTTS , we used only the speech domain datasets. The experimental results, as shown in Table 7, demonstrate that the Q2D2 model with only 53 tokens outperforms DAC and Ecodec models with one to nine quantizers and 100 to 900 tokens, on classification accuracy (except DAC with 9 quantizers at ravdess dataset). Also, the Q2D2 model achieves higher results in three of the four datasets compared to the WavTokenizer with 75 tokens.

Table 7: The semantic representation (speech) evaluation of various codec models on ARCH Benchmark in terms of classification accuracy. Nq represents the number of quantizers.

| Model | Nq ↓ | token/s ↓ | RAVDESS ↑ | SLURP ↑ | EMOVO ↑ | AM ↑ |
|---|---|---|---|---|---|---|
| DAC | 9 | 900 | **0.3750** | 0.0779 | 0.2363 | 0.6926 |
| Encodec | 8 | 600 | 0.2881 | 0.0636 | 0.2261 | 0.4388 |
| DAC | 4 | 400 | 0.3194 | 0.0782 | 0.2346 | 0.6838 |
| Encodec | 4 | 300 | 0.2951 | 0.0660 | 0.2193 | 0.4301 |
| Encodec | 2 | 150 | 0.2743 | 0.0627 | 0.2193 | 0.3649 |
| DAC | 1 | 100 | 0.2500 | 0.0713 | 0.2278 | 0.6287 |
| WavTokenizer | 1 | 75 | 0.3255 | 0.0802 | **0.3163** | 0.6957 |
| Q2D2 | 1 | 53 | 0.3298 | **0.0885** | 0.2448 | **0.709** |

## 5 ABLATION STUDY

For Q2D2 ablation studies we used 585 hours of LibriTTS training dataset and LibriTTS-test-clean subset for reconstruction performance. Our goal in the ablation studies was to understand the design choices of Q2D2, focusing on three main factors: *Grid type*, *Dimension size* and *Number of quantization levels*.

**Grid type.** We compared rhombic, rectangular, and hexagonal tilings under matched conditions (Table 8). The results show the rhombic grid consistently outperforming the others across PESQ and STOI. We attribute this improvement to its higher packing efficiency (as elaborate in D). Packing

Table 5: The subjective reconstruction results using MUSHRA (comparative scoring of samples) of codec models on speech domain. Nq denotes the number of quantizers.

| Model | Bandwidth ↓ | Nq ↓ | token/s ↓ | LibriSpeech and LJSpeech ↑ |
|---|---|---|---|---|
| GT | – | – | – | 98.08±1.47 |
| DAC | 9.0 kbps | 9 | 900 | 92.64±3.83 |
| Encodec | 6.0 kbps | 8 | 600 | 94.41±4.99 |
| Q2D2 | 3.3 kbps | 1 | 166 | **98.05±2.25** |
| WavTokenizer | 0.9 kbps | 1 | 75 | 94.83±2.63 |
| Q2D2 | 1 kbps | 1 | 53 | 94.68±3.05 |

Table 6: The Subjective Evaluations of various acoustic codec models on the LibriSpeech-test-clean and LJSpeech sets. GT denotes ground truth waveforms.

| Model | Bandwidth ↓ | Nq ↓ | CMOS-Q ↑ | CMOS-P ↑ |
|---|---|---|---|---|
| GT | – | – | 0.50 | 0.20 |
| WavTokenizer | 0.9 kbps | 1 | -0.30 | -0.50 |
| Q2D2 | 1 kbps | 1 | 0.00 | 0.00 |

efficiency determines how densely the quantization cells tile the 2-D embedding space higher packing efficiency yields more uniform latent-space coverage and lower quantization error for a fixed number of levels.

**Dimension size.** This ablation examines the impact of varying number of latent feature dimensions while fixing the bitrate at 1 kbps. As shown in Table 9, the 6-dimension configuration $[7, 7, 7, 7, 7, 7]$ achieves the best overall performance, yielding the highest UTMOS, PESQ, and STOI. Increasing to 8 dimensions with lower resolution or reducing to 4 dimensions with higher resolution degrades reconstruction quality, indicating insufficient feature representation. These results suggest that a moderate dimensionality offers the optimal trade-off between compactness and representational capacity.

Table 8: Impact of grid type on reconstruction performance.

| Model | Grid Type | UTMOS ↑ | PESQ ↑ | STOI ↑ | V/UV F1 ↑ |
|---|---|---|---|---|---|
| Q2D2 | Rhombic | **4.0312** | **2.3995** | **0.9152** | **0.9395** |
| Q2D2 | Rectangle | 4.0108 | 2.2909 | 0.9074 | 0.9370 |
| Q2D2 | Hexagon | 4.0093 | 2.2862 | 0.9072 | 0.9362 |

Table 9: Impact of dimension size on reconstruction metrics.

| Model | Grid size | Dimensions | Bandwidth ↓ | UTMOS ↑ | PESQ ↑ | STOI ↑ |
|---|---|---|---|---|---|---|
| Q2D2 | [5,5,5,5,5,5,3,3] | 8 | 1 kbps | 3.8112 | 2.092 | 0.8956 |
| Q2D2 | [7,7,7,7,7,7] | 6 | 1 kbps | **4.0312** | **2.3995** | **0.9152** |
| Q2D2 | [19,19,19,19] | 4 | 1 kbps | 3.7789 | 2.018 | 0.8951 |

**Grid quantization levels.** This ablation examines the effect of varying the number of quantization levels $l_i$ assigned to each dimension in the grid. Changing the level $l_i$ adjusts the resolution of the 2D grids and thus the overall bitrate of the codec. Configurations ranged from large resolutions $[11, 11, \dots]$ with high bandwidth (25.0 kbps) to smaller resolutions $[7, 7, \dots]$ at very low bandwidth (1 kbps). A minimum of $l_i \geq 7$ was enforced, as lower values had produced inadequate results in prior studies. The results shown in Table 10 illustrate a clear trade-off between bitrate, codebook utilization and reconstruction quality: larger grids provide finer quantization and higher UTMOS, PESQ, and STOI, but with reduced codebook utilization, while smaller grids achieve excellent utilization and lower bandwidth at the cost of moderate quality degradation.

Table 10: Impact of grid quantization levels on utilization and reconstruction metrics. Pair utilization reflects the usage of grid for all pairs. Codebook utilization rate reflects codebook's usage efficiency.

| Model | Grid Levels | Bandwidth ↓ | Pair Utilization ↑ | Codebook Utilization ↑ | UTMOS ↑ | PESQ ↑ | STOI ↑ |
|---|---|---|---|---|---|---|---|
| Q2D2 | [11,11,11,11,11,11] | 25.0 kbps | 100% | 72.11% | 4.0288 | **4.2832** | **0.9924** |
| Q2D2 | [9,9,9,9,9,9] | 9.5 kbps | 100% | 97.54% | 4.0002 | 3.9043 | 0.9747 |
| Q2D2 | [9,9,9,9,7,7] | 6.9 kbps | 100% | 99.42% | 4.0496 | 3.6975 | 0.9640 |
| Q2D2 | [9,9,7,7,7,7] | 3.3 kbps | 100% | **99.47%** | **4.0786** | 3.3870 | 0.9565 |
| Q2D2 | [7,7,7,7,7,7] | 1 kbps | 100% | 92.18% | 4.0312 | 2.3995 | 0.9152 |

Other ablation studies was preformed during the development process including spread factor $e_i$ and bounding impacts, STE substitutes, projections type and more as described in Appendix D. Overall, the ablations studies confirm that Q2D2's space design provides flexible trade-offs between bitrate, utilization, and reconstruction quality, with rhombic grids, moderate grid sizes, and balanced dimension counts yielding the most consistent gains.

## 6 CONCLUSION

In this work, we introduced **Q2D2**, a geometry-aware, two-dimensional quantizer capable of efficiently quantizing speech at bandwidths of 1kbps, 3.3kbps, and 6.9kbps. The audio and music domains will be studied in future work. Compared to SOTA codec models Q2D2 achieves high subjective reconstruction quality, while maintaining high codebook utilization, which preserves rich semantic information even under extreme compression. These findings suggest that Q2D2 design can serve as a powerful alternative to conventional scalar or vector quantization, capturing correlations across features more effectively.

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

## A  WAVTOKENIZER FRAMEWORK

Our implementation builds directly on the open-source WavTokenizer framework (Ji et al., 2024b). Unless otherwise specified, all components, training configurations, and data-processing pipelines follow the original WavTokenizer codebase. In our experiments, we modify **only the quantization layer**, replacing the default residual vector quantization modules with our proposed Q2D2 quantizer. All other system components remain unchanged to ensure a fair and controlled comparison. This design allows us to isolate the effect of the quantization scheme itself, enabling a direct assessment of how different grid types and quantization structures impact reconstruction quality.

## B  THE ARCH BENCHMARK

The ARCH benchmark comprises twelve datasets within the speech, music, audio domain. Emotional Speech and Song (RAVDESS) (Livingstone & Russo, 2012), Audio-MNIST (AM) (Becker, 2024), Spoken Language Understanding Resource Package (SLURP) (Bastianelli et al., 2020), and EMOVO dataset (Costantini et al., 2014) assess performance in the Speech domain. ESC-50 (Piczak, 2015), US8K (Salamon et al., 2014), FSD50K (Fonseca et al., 2021), and VIVAE (Holz et al., 2022) assess performance on Acoustic Events. FMA (Defferrard et al., 2016), MTT (Law et al., 2009), IR-MAS (Bosch et al., 2012), and MS-DB (Rafii et al., 2017) assess performance in the Music domain.

## C  SUBJECTIVE EVALUATIONS

For the subjective evaluations, we follow the MUSHRA protocol (ITU-R, 2001), using both a hidden reference and a low anchor. Annotators were asked to rate the perceptual quality of the provided samples from the test set in a range between 1 to 100.

For CMOS-Q and CMOS-P evaluations, we choose samples from the LibriSpeech test-set and LJSpeech for the subjective evaluation. We analyze the CMOS in two aspects: CMOS-Q (quality, clarity and high-frequency details), CMOS-P (Speech rate, pauses, and pitch). We instruct the testers to focus on the aspect in question and ignore the other aspect when scoring the aspect being considered.

## D  MORE ABLATION STUDIES

In our ablation experiments we asses more parameter and design choices as follow:

**Projections.** We further examined the role of input and output projection layers in shaping the quantization space. On the input side, we experimented with three alternatives: a simple linear projection, a linear layer followed by LayerNorm, and a linear layer followed by a $\tanh$ nonlinearity to bound the output within $[-1, 1]$. For the output stage, we tested both a linear projection back to the original dimensionality and an identity mapping. Across these variants, we observed that the bounded $\tanh$ projection provided the most stable training and best reconstruction quality, while purely linear or LayerNorm-based projections led to weaker performance and reduced robustness, as shown in shown in Table 11. This indicates that constraining the pre-quantization representation through a nonlinearity is crucial, and that the $\tanh$-based projection is the most effective choice in our framework.

Table 11: Ablation on input projection layers tested on the LibriTTS-test-clean dataset. Input projection reflects the diffrent projection strategies before quantization, including a simple linear projection and a linear projection followed by a $\tanh$ nonlinearity.

| Model | Grid Type | Grid Levels | Bandwidth ↓ | Input Projection | UTMOS ↑ | PESQ ↑ | STOI ↑ |
|---|---|---|---|---|---|---|---|
| Q2D2 | Hexagon | [7,7,7,7,7,7] | 1 kbps | linear projection | 3.2044 | 1.5513 | 0.8469 |
| Q2D2 | Hexagon | [7,7,7,7,7,7] | 1 kbps | linear projection + tanh | 4.0093 | 2.2862 | 0.9072 |

**Spread factor and bounding factor.** In our design, two parameters are closely related: the bounding factor, which restricts the feature dimension to the interval $[-\frac{l_i}{2}, \frac{l_i}{2}]$, and the grid spread factor $e_i = \frac{l_i-1}{2}$, which defines the spatial extent of the grid. Throughout development we experimented with alternative formulations of these factors, but in every case the result was degraded reconstruction quality and reduced codebook utilization. This indicates that the chosen formulation of spread and bounding factors is not only consistent but also critical for stable training and effective quantization.

**Hexagon grid offset.** We conducted an ablation study to investigate the correct offset needed in the $x$-axis of the hexagonal grid. In our implementation, every other row is shifted by $\pm\frac{dx}{4}$, where $dx$ is the horizontal spacing between points. This offset is essential for producing a true hexagonal tiling: without it, the grid degenerates into a rectangular lattice, and the geometric advantages of hexagonal packing are lost. The reason for the $\frac{dx}{4}$ factor comes from the geometry of equilateral triangles: a perfect hexagonal lattice can be constructed by stacking rows of points such that the centers of

adjacent hexagons align. Given horizontal spacing $dx$ and vertical spacing $dy = \frac{\sqrt{3}}{2} dx$, each odd row must be shifted by half the horizontal distance between neighboring points, i.e. $\frac{1}{2} \cdot \frac{dx}{2} = \frac{dx}{4}$. This guarantees that each point has six equidistant neighbors, forming the canonical hexagonal tessellation. We experimented with alternative offsets, including no shift, $\pm \frac{dx}{2}$, and arbitrary fractions, but found that only the $\pm \frac{dx}{4}$ offset consistently yielded a uniform hexagonal arrangement with correct neighbor relationships. Incorrect offsets resulted in uneven quantization densities and degraded reconstruction quality, whereas the $\pm \frac{dx}{4}$ rule provided the expected tessellation and stable performance. Results are shown in Table 12.

Table 12: Ablation on Q2D2 hexagonal grids with different offsets tested on the LibriTTS-test-clean dataset. Offset represents the $x$ construction offsets in the grid.

| Model | Grid Type | Grid Levels | Bandwidth ↓ | Offset | UTMOS ↑ | PESQ ↑ | STOI ↑ |
|-------|-----------|-------------|-------------|--------|---------|--------|--------|
| Q2D2 | Hexagon | [7,7,7,7,7,7] | 1 kbps | $\frac{dx}{2}$ | 3.8865 | 2.0825 | 0.8978 |
| Q2D2 | Hexagon | [7,7,7,7,7,7] | 1 kbps | $\frac{dx}{4}$ | 4.0093 | 2.2862 | 0.9072 |

**Gradient propagation.** We conducted an ablation study to investigate different strategies for propagating gradients through the quantization step. Specifically, we compared the straight-through estimator (STE), a rotation-based gradient trick, and several additional variations designed to stabilize training and improve codebook usage. Our experiments consistently showed that while alternative gradient tricks can introduce diversity, they often lead to unstable training dynamics or degraded performance. In contrast, the STE provided the most reliable optimization behavior, yielding stable convergence. This suggests that despite its simplicity, STE remains the most effective choice for gradient propagation in our framework.

**Gird packing efficiency.** Packing efficiency measures how densely quantization cells can tile the 2-D embedding space. Higher packing efficiency means that a larger fraction of the space is actually covered by usable cells, producing more uniform latent-space coverage and reducing quantization error for a fixed number of levels. Rhombic tiling provides higher packing efficiency than rectangular grids because its oblique basis yields a more isotropic, space-filling partition with less directional bias. While hexagonal cells are optimal for circle packing, they are less aligned with linear projection quantizers; rhombic grids better match the latent feature distribution, improving space utilization and reducing quantization distortion. This leads directly to the observed improvements in PESQ and STOI.

**Mutual-information.** To analyze the dependency structure induced by Q2D2, we compute mutual-information (MI) between the two coordinates of each 2D pair before and after quantization. The pre-quantization MI is nearly zero, indicating that the linear projections produce statistically independent components. After quantization, MI increases substantially, showing that the 2D grid introduces structured dependencies and compresses each pair onto a lower-dimensional manifold. This behavior helps explain patterns in codebook usage and the reconstruction differences observed across grid types. Resualts are shown in Figre 4.

# E   FUTURE WORK

Beyond two-dimensional grids, a key direction is extending to three-dimensional quantization geometries and systematically exploring new grid structures, including simplex tilings, higher-order polytopes, and mixed-dimensional partitions. Such designs may better capture correlations across latent features and provide richer representational capacity, enabling more efficient neural audio compression. Moreover, Q2D2 has not yet been extensively trained or evaluated on music and diverse audio domains, where its structured quantization could also deliver strong reconstruction quality. In terms of evaluation, a primary avenue for future work is to test our model on downstream tasks, particularly Text-to-Speech (TTS).

# F   RECONSTRUCTION SPEED

We evaluate the reconstruction speed of Q2D2 and WavTokenizer on a single NVIDIA RTX6000 48G GPU on the LibriTTS test-clean dataset. We calculate the real-time factor (RTF) by dividing

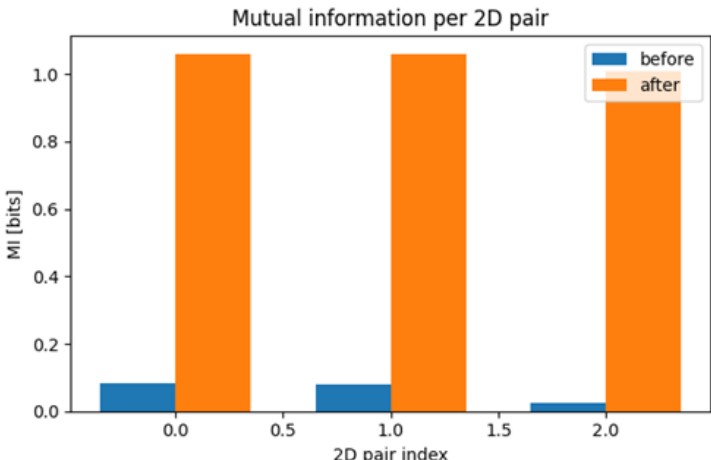

Figure 4: Mutual information between the two coordinates of each 2D pair before and after quantization. Pre-quantization MI is near zero, indicating independent components, while post-quantization MI increases significantly, showing that the 2D grid introduces structured dependencies.

the total reconstruction time by the duration of the generated audio. The results are shown in Table 13. These results demonstrate the high reconstruction efficiency of Q2D2.

Table 13: Reconstruction speed (measured by RTF) of different codec models on reconstruction on the LibriTTS-test-clean dataset. RTF is computed by dividing the total reconstruction time by the duration of the generated audio.

| Model | RTF |
|---|---|
| WavTokenizer | 0.0032 |
| Q2D2 | 0.0039 |

## G  USE OF LARGE LANGUAGE MODELS

In preparing this work, we employed large language models (LLMs) as auxiliary tools to streamline the research process. LLMs were helpful in polishing and clarifying writing, and improving readability. We also used LLMs for retrieval and discovery tasks, such as identifying relevant related work, create cites of papers and create figures or tables in the paper. Importantly, all technical contributions, experiments, and analyses remain the our original work.

