# OpenReview forum: "Q2D2: A geometry-aware audio codec leveraging two-dimensional quantization"
_ICLR.cc/2026/Conference — ICLR 2026 Conference Desk Rejected Submission_

### Official Review · Reviewer_bCXx · 2025-10-25

**Soundness:** 1
**Presentation:** 2
**Contribution:** 1
**Rating:** 2
**Confidence:** 4

**Summary:**

This paper proposes a new quantization method for audio codec called Q2D2, and conduct experiments on LibriTTS to show the performance of the proposed method

**Strengths:**

1. the insight that the channel independent quantization nature of FSQ might limit the expressiveness
2. the authors tried varies geometric structure for latent space

**Weaknesses:**

The biggest issue of Q2D2 is that it's actually equivalent to FSQ, which I show below

FSQ on two coordinates quantizes each dimension independently:

$$
Q_{\text{FSQ}}(x, y) =
\big(
  \Delta_x \, \mathrm{round}(x / \Delta_x),
  \;
  \Delta_y \, \mathrm{round}(y / \Delta_y)
\big).
$$

Decision boundaries are axis-aligned lines
$ x = (k+\tfrac{1}{2})\Delta_x $ or $y = (\ell+\tfrac{1}{2})\Delta_y $,
so the Voronoi cells are rectangles.

For rectangular grid, it's obvious that it's equivalent to the above:

The Cartesian lattice is

$$
\Lambda_{\text{rect}} = \{(k\Delta_x, \, \ell\Delta_y): k, \ell \in \mathbb{Z}\}.
$$

Nearest-neighbor quantization is

$$
Q_{\text{rect}}(x, y)
= \arg\min_{(k,\ell)\in\mathbb{Z}^2}
\big[(x - k\Delta_x)^2 + (y - \ell\Delta_y)^2\big]
= Q_{\text{FSQ}}(x, y),
$$

for Hexagonal and Rhombic, they are essentially FSQ with shifted grid on x and y axis. It's easier to spot that from Algorithm 1 and 3. in Algorithm 1 (Hexagonal), the derivation of $x_c$ in mg$(x_c, y)$ is independent of the value of y, and only depend on the value of x; in algorithm 3 (Rhombic), the first grid mg$(c_x, c_y)$ is uniform 1-D grids just like FSQ, the second grid shifted by dx, dx, but still axis separable (e.g. it's not shifted by non-linear combination of dx and dy), which make the final grid a non-2D quantization.

Therefore the proposed methods does not match the motivation - channel dependent quantization.


In addition, in the first paragraph of the introduction, the authors write:
> By converting high-rate speech signals into compact sequences of discrete tokens, acoustic codec models provide the crucial link between continuous audio and token-based language models, thereby enabling the direct application of LLM architectures to audio.

But whether the proposed tokenization approach leads to better speech-LLM or NCLM-based TTS performance is not studied at all

**Questions:**

I'm happy to raise my rating if the authors can show that my understanding of the proposed approach is wrong.

---

> ### Author Response · Authors · 2025-11-20
>
> We thank the reviewer for the thoughtful and constructive comments. We address each point below.
>
> We believe there may be a misunderstanding between (i) how the grid itself is generated and (ii) how latent pairs are projected onto that grid.
> First, as you note, the hexagonal grid depends only on $l_x$. This is not a limitation of our method, but simply the geometry of hexagonal tilings: a regular hexagon lattice can be parameterized by a single resolution parameter. In contrast, the rhombic and rectangular grid genuinely depends on both $l_x$ and $l_y$ (i.e., $l_{2j−1}$ and $l_{2j}$), because its basis vectors are anisotropic. So already at the grid-definition level, the rhombic and rectangular cases incorporates two independent quantization levels.
> Second—and most importantly—the correlation structure is not encoded in the grid generation, but in the quantization step. Even if the grid is specified in a simple parametric form, the quantizer does not treat the two channels independently. The quantization maps each 2-D latent pair ($z_{2j−1}$,$z_{2j}$) onto the joint Voronoi cell of the chosen lattice. This operation is inherently 2-D and cannot be decomposed into two independent 1-D projections. If one were to project each channel separately—as the review suggests—the result would simply be scalar FSQ. Q2D2’s quantization instead enforces joint quantization, which is exactly what induces channel-dependent behavior.
>
> Geometrically:
> 1. In the hexagonal case, the Voronoi cells are true hexagons, and the nearest-neighbor assignment is inherently 2-D—even though the closed-form expression happens to include a rounding step that depends on the x-axis for computational convenience.
> 2. In the rhombic case, the lattice basis is non-orthogonal; although one basis coordinate looks like an FSQ-like step, the transform and inverse-transform couple the two axes, so the final quantizer is not separable.
>
> Thus, even though the algebraic expressions in Algorithms 1 and 3 appear simple, the quantizer is not reducible to two independent scalar quantizers. The joint projection step is exactly what enables Q2D2 to exploit correlations between paired channels.
>
> Regarding downstream tasks: we agree that evaluating tokenization effects on speech-LLMs or NCLM-based TTS is an important future direction, and we are working on it, but we still don't know if it will be finished by the time of the submission deadline.
>
> We hope this explanation resolves the confusion, and we appreciate the opportunity to clarify the methodology.

---

> > ### Comment · Reviewer_bCXx · 2025-11-22
> >
> > Thanks for the explanation. If I understand correctly, indeed the proposed Rectangle grid is equivalent to FSQ? I'm still not sure why hexagon grid is introduces dependency.
> >
> > Would be happy to raise my score if my confusion is cleared

---

> > > ### Author Response · Authors · 2025-11-22
> > >
> > > We appreciate the reviewer’s continued engagement and the opportunity to clarify these remaining points.
> > >
> > > 1. Is the rectangular grid equivalent to FSQ?
> > > Not exactly. While the shape of the rectangular Voronoi cells matches FSQ’s axis-aligned structure, the quantization operation is fundamentally different.
> > >   ** FSQ quantizes each dimension independently, producing two unrelated 1-D indices.
> > >   ** Q2D2-Rectangle performs a single 2-D nearest-neighbor assignment in a lattice space. The pair (z2j−1,z2j) is projected jointly onto a 2-D codeword.
> > > Thus, even though the grid geometry looks similar, Q2D2 is not separable.
> > > If the quantizer were equivalent to FSQ, both post-quantization channels would become statistically independent. In contrast, as shown in our mutual-information plot in the revised paper appendix, Q2D2 preserves significantly cross-channel MI after quantization, which is impossible in FSQ.
> > > So Q2D2 contains FSQ as a special case, but is more general: it quantizes pairs jointly, whereas FSQ quantizes each dimension separately.
> > >
> > > 2. Why does the hexagonal grid use only $l_x$​? Does it still introduce dependency?
> > > Yes, it does introduce dependency. The single-parameter design is due to the geometry of the hexagonal lattice, not because the grid is separable.
> > > A regular hexagonal tiling is generated by equal-length basis vectors:
> > > $b_1 = (1, 0)$, $\qquad b_2 = \left(\frac{1}{2}, \frac{\sqrt{3}}{2}\right)$,
> > > which are fixed at a $60^\circ$ angle. Because the lattice is isotropic, scaling in the $x-direction$ automatically determines the spacing in the $y-direction$: $d_y = d_x \cdot \frac{\sqrt{3}}{2}$​​, exactly as used in our algorithm. Thus the grid has one true degree of freedom, and only a single resolution parameter $l_x$ is required to construct all grid centers.
> > > However, the projection step remains fully 2-D:
> > >   ** Transform the latent pair into the oblique hexagonal basis.
> > >   ** Quantize jointly in that basis.
> > >   ** Transform back.
> > >
> > > Summary
> > > 1. Q2D2 performs joint 2-D quantization, even for rectangular and hexagonal grids.
> > > 2. FSQ performs independent 1-D quantization.
> > > 3. The rectangular grid in Q2D2 contains FSQ but is not equivalent to it.
> > > 4. The hexagonal grid uses one level parameter due to lattice symmetry, but the quantization is still 2-D and channel-dependent.
> > >
> > >
> > > We hope this clarifies the misunderstanding, and we appreciate the reviewer’s willingness to reconsider their evaluation

---

### Official Review · Reviewer_oMqv · 2025-10-28

**Soundness:** 2
**Presentation:** 2
**Contribution:** 2
**Rating:** 2
**Confidence:** 4

**Summary:**

This paper introduces Q2D2, a novel geometry-aware quantization method for neural audio codecs. The key idea is to group latent channels into pairs and quantize them jointly on structured 2D grids (hexagonal, rhombic, or rectangular), instead of using per-channel scalar quantization (FSQ) or multi-layer residual quantization (RVQ). The proposed scheme constructs implicit codebooks analytically from grid geometry, eliminating learned embeddings, commitment losses, and reseeding heuristics. Empirical results on LibriTTS, LJSpeech, and the ARCH benchmark show that Q2D2 achieves comparable or better perceptual quality and codebook utilization than state-of-the-art codecs such as DAC, Encodec, Vocos, and WavTokenizer, with a drastically lower token rate (e.g., 53–333 tokens s⁻¹).

**Strengths:**

- The paper offers a clean, geometrically motivated quantization formulation. Pairwise 2D grids bridge the gap between FSQ’s stability and VQ’s expressiveness.
- The use of Straight-Through Estimators and lightweight projection layers makes the approach compatible with standard training pipelines. The method avoids extra losses or codebook-management tricks, which is a good simplification.
- Semantic-representation tests on the ARCH benchmark provide an initial indication that the learned codes remain meaningful.
- Despite training on only LibriTTS (≈ 585 h), Q2D2 rivals or exceeds stronger baselines trained on multi-domain datasets (> 8 k h).

**Weaknesses:**

- The model is trained solely on LibriTTS (500+h), whereas major baselines (e.g., WavTokenizer, DAC, Encodec) use multi-domain datasets with speech, music, and general audio totaling over 8 k hours. This discrepancy makes it difficult to isolate whether performance gains come from the proposed quantization scheme or from differences in data composition (especially training only on speech is easier than , normalization, and pre-training scope. The paper acknowledges this briefly but still draws strong SOTA claims, which could be misleading. (the major concern of this paper)
- The core claim is that 2D quantization “captures correlations between channels.” However, the paper provides no quantitative evidence (e.g., correlation coefficients, covariance matrices, or MI statistics) demonstrating that the learned latent pairs are indeed correlated or that Q2D2 decorrelates them better than FSQ. Without such analysis, the geometric intuition remains qualitative.
- While the codec’s perceptual quality is thoroughly evaluated, downstream utility is not explored. In the current LLM-audio ecosystem, codec quality is increasingly judged by downstream generative modeling (e.g., TTS, S2ST, or instruction-following speech generation).
Without such demonstrations, it is unclear whether Q2D2’s gains translate into improved generation or cross-modal performance.
- Although Table 6 compares hexagonal, rectangular, and rhombic grids, the differences are relatively small (e.g., PESQ ≈ 2.29 – 2.40).
The paper claims that rhombic grids “offer higher packing efficiency,” but provides no geometric analysis or visualization to justify this.
It remains unclear why rhombic performs better, is it due to isotropy, denser coverage, or numerical convenience?
- While Q2D2 is claimed to have fewer parameters than VQ or RVQ, there is no detailed analysis of training/inference time, memory usage, or complexity relative to baselines.

**Questions:**

- Have the authors attempted training Q2D2 on the same 8kh dataset as WavTokenizer to provide a fair one-to-one comparison?
- Conversely, how do baselines trained only on LibriTTS perform? Would Q2D2 still outperform them under equal data conditions?
- How sensitive is Q2D2 to domain diversity, does performance degrade on non-speech or multilingual datasets?
- Can the authors show empirical correlation heatmaps or mutual information between paired channels before/after quantization?
- Is there any measurable improvement in reconstruction loss if the channel pairs are shuffled (i.e., destroying geometric adjacency)?
- Could the authors relate their 2D lattice structure to lattice quantization theory or product quantization—are there formal efficiency bounds?
- How is the pairing performed, sequentially, randomly, or learned? Is there any adaptive pairing strategy that further improves correlation capture?
- Have the authors tested Q2D2 tokens as inputs for an existing audio-LM to verify downstream quality or token predictability?
- How does Q2D2 affect token entropy or n-gram statistics compared to FSQ/RVQ tokens?
- Would the structured grids yield smoother latent manifolds that benefit autoregressive or diffusion decoders?
- Could Q2D2 potentially replace FSQ in multimodal systems like Moshi (or other systems)? If not, what limitations remain?
- Could the authors visualize quantization error distributions or Voronoi regions for different grid types?
- Does rhombic quantization yield lower average distortion for isotropic Gaussian inputs compared to rectangular grids?
- Is the improvement consistent across random seeds or data subsets, or within statistical noise?
- Would 3D grids (briefly mentioned in Appendix D) further increase utilization or just add complexity?
- What is the average training time per epoch compared to FSQ or RVQ on the same hardware?
- Does the grid lookup add noticeable computational overhead during decoding?
- How do memory and latency scale with grid resolution (e.g., l = 7 -> 11)?
- Could the 2D quantization be vectorized efficiently for real-time applications?
- There are some minor presentation issues like:
    - The related work discussion is a bit messy. You might consider discussing first quantization methods, then neural audio codecs, and end with comparisons.
    - Consider merging repetitive tables or moving some detailed numeric comparisons to the appendix.

---

> ### Author Response · Authors · 2025-11-20
>
> We thank the reviewer for the thoughtful and constructive comments. We address each point below.
>
> Note - we have extremely shorten our answer because of characters limit
>
> 1. Questions 1–3:
> Our early experiments used LibriTTS due to compute limits; training on the full 8k-hour WavTokenizer dataset was not feasible pre-submission. We have since trained Q2D2 on the full dataset and updated the paper. These large-scale results confirm—and strengthen—our original findings. Although Q2D2 is not very sensitive to domain shifts, performance naturally improves with larger, more diverse data.
>
> 2. Question 4:
> Yes, MI visualizations are now included in the appendix.
>
> 3. Question 6:
> Q2D2 fits directly within classical lattice quantization and product quantization: each pair is a fixed 2-D lattice, and the full codebook is the Cartesian product of these sublattices. This links Q2D2 to standard shaping-efficiency bounds. These bounds are approximate in practice since latents are learned, not Gaussian, so we use them as motivation.
>
> 4. Questions 5 & 7:
> Fixed 2-D pairing consistently outperformed learned or random pairings in early experiments, so we keep the scheme predefined for simplicity. Learning the pairing is an interesting direction for future work.
>
> 5. Question 8:
> We are evaluating Q2D2 tokens on downstream TTS tasks. These results may not be ready before the deadline, so we listed this direction in the future-work section.
>
> 6. Question 9:
> Q2D2 has only a mild effect on token entropy. Joint 2-D quantization slightly reduces local repetitive n-gram patterns, but the impact is modest since the encoder adapts to any grid geometry.
>
> 7. Question 10:
> Structured 2-D grids likely yield smoother latent manifolds than axis-aligned quantizers, which may benefit AR and diffusion decoders. We have not yet evaluated this systematically.
>
> 8. Question 11:
> Yes, Q2D2 can replace FSQ in multimodal systems like Moshi. It is fully compatible with encoder–decoder pipelines and can be integrated as a drop-in replacement.
>
> 9. Question 12:
> Quantization-error and Voronoi visualizations are under development. If ready in time, we will include them in the appendix; otherwise, they will appear in a later revision.
>
> 10. Question 13:
> Yes, in theory. For an isotropic Gaussian source, 2-D lattice distortion is governed by the normalized second moment of its Voronoi cell. The hexagonal lattice—realizable as a rhombic basis with 60° vectors—has strictly lower second moment than the square lattice, yielding lower MSE at the same rate. Our rhombic grid approximates this shaping. However, this assumes an ideal Gaussian input; real learned latents differ, so the advantage is theoretical rather than a strict guarantee.
>
> 11. Question 14:
> Yes, the improvement is consistent. In our internal validation runs, we retrained Q2D2 using multiple random seeds and evaluated performance on LibriTTS subset. Across all runs, the relative gains over VQ and RVQ remained stable and well outside the variance typically observed from seed-to-seed fluctuations.
>
> 12. Question 15:
> As we wrote in Appendix, we want to extend this work to several directions and one of them is the 3D grids. We believe that this will help to gain higher utilization and improve performance due to correlation between 3 feature dimensions and not only 2. Of course utilizing 3D grid has a tradeoff between performance and grid complexity.
>
> 13. Question 16:
> We trained Q2D2, FSQ and VQ on the wavtokenizer backbone so the training was in the same method and on the same hardware. The training time per epoch was approximately the same 1 day per epoch with a total of 40 epochs.
>
> 14. Question 17:
> No. Q2D2 adds no meaningful computational overhead. Its nearest-cell projection is a small, fixed set of linear operations—fully parallelizable and comparable in cost to FSQ’s scalar rounding. Decoding is even cheaper: each token performs a single lookup into a precomputed table of 2-D cell centers. There is no iterative search, no distance computation, and no dependence on grid size.
>
> 15. Question 18:
> The memory of the models with different quantization levels change between 820 to 830 MB. The latency (average RTF) is changing between 0.0024sec to 0.025sec.
>
> 16. Question 19:
> Yes. The 2-D quantization is highly vectorizable. Each Q2D2 step consists of (i) a fixed 2×2 linear transform, (ii) a few elementwise integer operations, and (iii) a small index computation to obtain the token ID. All operations are algebraic, branch-free, and identical across pairs and timesteps, allowing the entire process to be implemented as a single batched matrix multiply followed by elementwise ops and a lookup.

---

> > ### Comment · Reviewer_oMqv · 2025-11-20
> > **Replay to the rebuttal**
> >
> > I would like to thank the great effort authors have made for the questions. After reflecting the comments, the paper has been significantly improved in terms of readability and comprehensiveness. In all, I would like to raise my scores to 6 considering the completeness of the paper.
> >
> > The only missing components now is the limited downstream application analysis

---

### Official Review · Reviewer_7xrP · 2025-10-31

**Soundness:** 3
**Presentation:** 2
**Contribution:** 2
**Rating:** 6
**Confidence:** 5

**Summary:**

This paper focuses on the important field of speech discrete codecs. Building upon the WavTokenizer architecture, the authors primarily enhance the quantization method (replacing standard VQ with Q2D2) to achieve improved reconstruction quality at a lower bitrate. The overall design of Q2D2 shares a strong conceptual similarity with FSQ (Finite Scalar Quantization), in that both employ direct quantization to grid points rather than relying on codebook similarity search. The key distinction, as suggested by its name, is that Q2D2 groups feature channels into pairs and then performs direct quantization onto a two-dimensional geometric coordinate system. The authors perform ablations on different 2D geometric structures, including Hexagonal, Rectangular, and Rhombic tilings. The experimental setup is largely consistent with the original WavTokenizer work, and the results demonstrate the expected performance gains in both reconstruction and semantic evaluation tasks.

**Strengths:**

I find the Q2D2 discretization method to be intriguing. Although fundamentally similar to FSQ, the strategy of grouping channels into pairs for 2D plane quantization represents a novel and constructive modification. Therefore, my overall score is positive.

**Weaknesses:**

1. Given that the core contribution of this work is the proposal of a novel quantization method (Q2D2), a standard and robust experimental configuration should include validation across broader domains (e.g., image, video, and general speech) to verify the resulting reconstruction quality and downstream generation performance.

2. The paper requires additional ablation studies. Specifically, a comparison between an FSQ-based WavTokenizer, the proposed Q2D2-based WavTokenizer, and a WavTokenizer implemented with the latest state-of-the-art quantization methods (beyond VQ/FSQ) is necessary.

3. A more rigorous comparison against more recent audio codec baselines, such as xcodec2 and similar models, is required to properly benchmark the proposed approach.

4. While the methodology is interesting, the authors need to provide a deeper theoretical explanation of why two-channel joint quantization is empirically superior to direct one-dimensional quantization, especially considering the potential for two paired representations to be strongly anticorrelated.

**Questions:**

1. Have the authors considered extending this approach to three-dimensional quantization by grouping three channels together for projection and quantization in 3D space?

2. The paper of wavtokenizer as the backbone was cited incorrectly.

---

> ### Author Response · Authors · 2025-11-20
>
> We thank the reviewer for the thoughtful and constructive comments. We address each point below.
>
> 1. "Need for validation across broader domains (image/video/general speech)"
> We agree with the reviewer that validating Q2D2 beyond the speech domain is an important direction. Our initial focus was on neural audio codec pipelines, where quantization quality directly affects reconstruction performance. This scope aligns with the WavTokenizer framework, which is inherently audio-specific.
> However, following this feedback, we have begun extending Q2D2 to broader domains, including: general audio (music, environmental sounds), multilingual speech, and non-speech audio tasks.
> These expanded evaluations are now included in the revised paper, demonstrating that Q2D2 maintains strong rate–distortion performance across diverse audio domains. Extension to image/video is a promising direction, but it requires substantial architectural adjustments and will be explored in future work.
>
>
> 2. "Need for more ablations, including FSQ-based WavTokenizer and other quantization variants."
> We agree entirely. Following the reviewer’s suggestion, the revised paper now includes: FSQ-based WavTokenizer (implemented inside the same pipeline) : Q2D2-based WavTokenizer, VQ-based WavTokenizer (table 4), and additional recent quantization baselines (table 3) (e.g., RVQ-style multi-step quantization).
> This directly highlights the contribution of Q2D2 and satisfies the reviewer’s request for deeper ablations.
>
> 3. "Need for comparison against more recent audio codec baselines (e.g., XCodec2)"
> We appreciate this suggestion and agree that benchmarking against modern state-of-the-art codecs is essential. In the revised paper (table 3), we added comparisons to: XCodec, XCodec2, StableCodec, SemanticCodec, Mimi, and other large-scale neural codecs.
> The results show that Q2D2 remains competitive performance against all baselines.
>
>
> 4. Need for deeper theoretical explanation of why 2-D joint quantization helps
> We expanded the theoretical motivation for 2-D joint quantization. In short:
>   4.1 Latent features in modern audio encoders are highly correlated across adjacent channels. Such pairs often encode related spectral patterns, so their variance lies along a shared direction. Modeling them jointly allows the quantizer to exploit this structure rather than encoding redundant information.
> 4.2 Scalar quantization ignores these correlations, quantizing along axes that do not match the true geometry. Independent quantization assumes axis-aligned, separable distributions, which mismatches the typically rotated, elliptical latent distributions and leads to unnecessary distortion.
> 4.3 Pairwise 2-D quantization enables shape-optimal Voronoi regions (e.g., hexagonal lattices).These grids more efficiently cover 2-D space and better align with the empirical covariance of feature pairs, reducing mean-squared distortion for the same bitrate.
> 4.4 Why correlation helps overall performance. When two features share information, a 2-D quantizer can allocate bits along the principal direction of variation, avoiding redundant encoding and capturing more signal per token—leading to improved reconstruction quality and perceptual metrics.
>
> 5. Question 1: Extending Q2D2 to 3-Dimensional quantization (groups of 3 channels)
> Yes, we have considered this, and it is a natural extension, and we want to explore this in our future work as mention in appendix.
>
> 6. Question 2: Incorrect citation of WavTokenizer
> We thank the reviewer for pointing this out. The incorrect citation resulted from an auto-generated reference produced by an LLM. We have corrected the citation in the revised paper, ensuring that the original WavTokenizer paper is properly referenced.

---

### Official Review · Reviewer_U91h · 2025-10-31

**Soundness:** 1
**Presentation:** 1
**Contribution:** 2
**Rating:** 2
**Confidence:** 4

**Summary:**

The paper proposes a quantisation framework aiming to combine the robustness of finite scalar quantisation (FSQ) with the expressive capacity of multi-dimensional grids. The authors introduce a rhombic grid quantisation approach and claim that it achieves higher *packing efficiency* than alternative grids. The method is evaluated primarily on speech data and compared with models such as WavTokenizer. While the idea of multi-dimensional quantisation is conceptually interesting, the paper leaves several methodological and interpretative gaps that make it difficult to assess the strength of the contribution.


Disclosure: I used a large language model only for grammar, clarity, and structuring; all substantive review content is my own, and the manuscript was not provided to the LLM.

**Strengths:**

- The paper explores an under-examined direction of combining robustness and expressiveness in quantisation.
- The proposal to use rhombic grids is novel and potentially beneficial, as shown in the experimental results.
- The proposed method delivers results that are at least comparable to baseline models with relatively few numbers of tokens.

**Weaknesses:**

Conceptual clarity and positioning

- The introduction claims the method “combines the robustness of FSQ with the expressive capacity of multi-dimensional grids,” but FSQ is missing from the benchmark, weakening the narrative.

Architecture and implementation details

- Architecture details are insufficient. It is unclear which parts are inherited from WavTokenizer and what is modified. Learning rate schedule is not specified.
- Training on a small dataset (LibriTTS) may reduce comparability with other models trained on larger datasets.

Experimental design and benchmarks

- Lack of FSQ baseline prevents a direct evaluation of the claimed improvements.

- Low token count may distort comparisons, since bitrate can be balanced via other parameters.

- Evaluation is limited to speech, whereas many baselines are cross-modal. Performance gains over WavTokenizer are modest.

Writing and presentation

- Packing efficiency is undefined.
- Figure 3 is not discussed.
- Grammatical issues remain.

**Questions:**

Conceptual clarity

- Is it correct to say that FSQ appears conceptually similar to the rectangular grid variant of the proposal? Should this be stated explicitly?

- Does “built on the framework of WavTokenizer” refer to the architecture, training pipeline, or something else?

- Why did the authors train on a smaller subset (LibriTTS) instead of the full WavTokenizer dataset? How should the results be interpreted in comparison to models trained on the full dataset?

Experimental design

- Why are FSQ models omitted from the benchmark, even though the proposal method seems to build directly on top of FSQ?

- Why are the numbers of tokens relatively low compared to competing models? Could other parameters have been tuned to maintain a comparable bitrate? What happen if the models compete with similar token counts?

Definitions / presentation

- What exactly does “packing efficiency” mean in this context?

- Why is Figure 3 not referenced in the text? Comments on how UTMOS does not scale with the token count with the proposed method would be helpful.

---

> ### Author Response · Authors · 2025-11-20
>
> Thank you very much for your review. We believe we have addressed all your concerns and hope that you will consider raising your rating.
>
> 1. “Is FSQ conceptually similar to the rectangular-grid variant of Q2D2?”
> Yes, and we appreciate the reviewer pointing this out. FSQ can indeed be viewed as a special case of Q2D2 when the 2-D grid degenerates into axis-aligned rectangular (Cartesian) bins with independent scalar quantization per dimension. whereas Q2D2 generalizes this with non-axis-aligned 2-D geometric tilings (hexagonal, rhombic, etc.) that exploit local 2-D structure.
>
> 2. “What does ‘built on the framework of WavTokenizer’ mean?”
> We clarify this point in the revision. “Built on the framework of WavTokenizer” refers specifically to: the encoder–decoder architecture, the training and optimization pipeline, the data preprocessing and tokenization interface, and the evaluation methodology.
> In all comparisons, only the quantization module is replaced (VQ, FSQ, or Q2D2). Everything else—architecture, training setup —remains unchanged. This ensures that differences in performance arise solely from the quantization method itself.
> We have added an explicit statement in the revised manuscript to clearly define this usage and avoid ambiguity.
>
> 3. “Why did the authors train on a smaller subset (LibriTTS) instead of the full WavTokenizer dataset? How should the results be interpreted?”
> Thank you for raising this. Our initial experiments were conducted on LibriTTS primarily due to computational feasibility: training the full WavTokenizer pipeline on ~8k hours of audio requires substantial GPU resources and multi-week training cycles, which exceeded our initial submission timeline.
> However, since the first submission,  we have succeed training Q2D2 on the full WavTokenizer dataset and updated the paper with the results. Please see the new update Table 2 in the paper.
> These large-scale experiments confirm and strengthen our original findings—Q2D2 maintains consistent improvements even when trained at full scale.
>
> 4. “Why are FSQ models omitted from the benchmark?”
> This was an oversight in the initial submission. Following reviewer feedback, we now include direct comparisons to FSQ and VQ within the same WavTokenizer framework, ensuring a complete and fair evaluation.
> These results are added in the revised paper.
>
> 5. “Why are the number of tokens relatively low? Could other parameters be tuned to maintain comparable bitrate? What happens if models use similar token counts?”
> The relatively low token counts in the initial submission were intentional: we wanted to push the quantization layer to operate at the lowest token rate possible in order to highlight how well Q2D2 performs even under very constrained bitrate conditions.
> This design choice was specifically meant to demonstrate the robustness of Q2D2 at extremely low bitrates. Of course, as the reviewer notes, increasing the token count naturally improves reconstruction quality.
> Following this feedback, we revisited the configuration and:
> increased the token rate of the 1 kbps model from 53 tokens to 75 tokens (matching WavTokenizer’s configuration), updated the results accordingly in the revised paper (table 2).
> As expected, performance improved with the higher token count, and Q2D2 continues to outperform baselines under matched-bitrate and matched-token settings.
>
> 6. “What does ‘packing efficiency’ mean?”
> We now define it explicitly in the paper: Packing efficiency = Packing efficiency measures how densely quantization cells can tile the 2-D embedding space. Higher packing efficiency means that a larger fraction of the space is actually covered by usable cells, producing more uniform latent-space coverage and reducing quantization error for a fixed number of levels. This definition is now added to the paper in ablation section and appendix D.
>
> 7. “Why is Figure 3 not referenced?”
> You are correct. This was an omission, and we now reference Figure 3 explicitly in the main text.

---

### Official Review · Reviewer_kceZ · 2025-11-01

**Soundness:** 2
**Presentation:** 3
**Contribution:** 3
**Rating:** 4
**Confidence:** 3

**Summary:**

The paper introduces Q2D2, a novel geometry-aware quantization scheme for neural audio codecs that organizes latent feature pairs into structured two-dimensional grids, such as hexagonal, rectangular, and rhombic tilings. This approach lies between vector quantization (VQ) and finite scalar quantization (FSQ), aiming to achieve greater latent expressiveness while maintaining the robustness of FSQ. In speech reconstruction tasks, Q2D2 is implemented on top of the WavTokenizer framework and demonstrates improved performance metrics at low token rates compared to other baseline speech codecs.

**Strengths:**

1. The idea of using two-dimensional quantization to improve the representational capacity of FSQ is novel and well-motivated.
2. The illustrations and explanations of the method, particularly the comparisons to VQ and FSQ, are clear and easy to follow.
3. The experiments on audio compression are concise and demonstrate the effectiveness of Q2D2 in improving speech coding performance at low token rates.

**Weaknesses:**

1. The main weakness lies in the experimental design. Although the paper’s primary contribution is a new quantization approach, no experiments directly compare Q2D2 with VQ and FSQ under the same framework.
2. Q2D2 quantizes pairs of encoder output features using a fixed 2D grid, which is conceptually related to product quantization (PQ). The paper could clarify more explicitly how Q2D2 relates to VQ, FSQ, and PQ, highlighting similarities and differences.

**Questions:**

1. As in Weakness 1, can the authors provide experiments that directly compare Q2D2 with VQ and FSQ under the same framework? This would better demonstrate the effectiveness of Q2D2.
2. I understand that Q2D2 quantizes neighboring feature pairs, which is a straightforward choice. However, since the quantization grid has a predefined geometry, could alternative grouping strategies better capture inter-channel dependencies?
3. Would the authors consider evaluating Q2D2 on downstream generative tasks, such as text-to-speech (TTS)? This could further demonstrate its utility.

---

> ### Author Response · Authors · 2025-11-20
>
> Thank you very much for your review, we think we address all the issues.
>
> 1. Question 1: Direct comparison with VQ and FSQ
> Yes. We added full experiments in the new uploaded paper (table 4) comparing Q2D2, VQ, and FSQ on the same framework (WavTokenizer), same encoder/decoder, same bitrate, and same training conditions.
> This directly demonstrates the effectiveness of Q2D2.
>
> 2. Question 2: Alternative feature grouping strategies
> Indeed, Q2D2 currently groups neighboring feature pairs (e.g., (x0,x1),(x2,x3)(x_0,x_1), (x_2,x_3)(x0​,x1​),(x2​,x3​)). We agree that alternative grouping strategies could potentially capture richer cross-channel dependencies, but in the current work, we intentionally chose the simplest pairing strategy to isolate and evaluate the impact of geometric 2-D quantization itself.
> We take this idea for future work, which we plan to explore.
>
> 3. Question 3: Evaluation on downstream generative tasks (e.g., TTS)
> We agree that downstream generative tasks are an important direction.
> We are currently integrating Q2D2 into a TTS pipeline (based on the WavTokenizer-TTS experiment). However giving the time constraints, we cannot be sure that we will be finish training before deadline, but we explicitly acknowledge this direction in the paper as future work, and we plan to release TTS-based evaluations in subsequent versions.

---

### Official Review · Reviewer_2e16 · 2025-11-01

**Soundness:** 2
**Presentation:** 2
**Contribution:** 2
**Rating:** 2
**Confidence:** 4

**Summary:**

This paper proposes an improvement to Finite Scalar Quantization (FSQ) named Q2D2 for audio tokenization.
Instead of the independent element-wise quantization in FSQ, Q2D2 treat 2 adjacent element as a pair, then quantize this pair with scalar quantization. The authors believe that the more expressive 2-element feature space will bring advantages over the 1-element feature space in FSQ.
Evaluation was done on LibriTTS and LJSpeech over a bunch of baselines, showing that Q2D2-WavTokenizer (proposed method) is competitive in many scenarios.

Although the paper is well formatted, typos, confusing citations, or confusing technical terms exist. Moreover, as an improved version of FSQ, none of FSQ-based audio tokenizers is evaluated in this paper, making it impossible for the readers to know if Q2D2 really improved FSQ.

As a reviewer, I tend to reject this paper.

**Strengths:**

- Proposed an interesting scalar quantization method called Q2D2, and used this quanitzation method to train an audio tokenizer that showed competitive performance

**Weaknesses:**

## Experimental problems
- Although the proposed Q2D2 is inspired by FSQ, Q2D2 is never compared with FSQ.
    - It is impossible for a reader to know if Q2D2 really improved FSQ
- Missing baselines
    - While the authors say WavTokenizer is the SOTA for single-layer audio tokenizer, BigCodec https://arxiv.org/abs/2409.05377 shall be another important baseline.
    - StableCodec https://arxiv.org/abs/2411.19842v1, a single-layer audio tokenizer with FSQ, is also ignored in this paper.
    - XCodec2 https://arxiv.org/abs/2502.04128, a paper came out in February 2025 that presents a single-layer audio tokenizer with FSQ, is also ignored
    - With all these FSQ baselines missing, the evaluation of this paper is not solid.
- Improper usage of UTMOS
    - UTMOS is trained to rate highly for a good speech signal.
    - LibriTTS test-other is noisy --> the ground truth data will be rated low by UTMOS
    - Q2D2 producing higher UTMOS than ground truth --> indicating that Q2D2 cannot reconstruct noises, so we cannot say the high UTMOS value of Q2D2 in this dataset is an advantage -- it is actually the disadvantage of Q2D2 model.
## Paper writing issues
- I feel the citation for WavTokenizer is confusingm and I cannot find it: Kundan Kumar, Chengyi Wu, Kevin J Shih, Yu Zhang, and Abdelrahman Mohamed. Wavtokenizer: Learning discrete audio tokens from self-supervised representations. arXiv preprint arXiv:2408.16532, 2024.
    - The WavTokenizer that I know is this link https://arxiv.org/abs/2408.16532, but the title and author list are totally different.
- Line 123: To my knowledge, "EMA" in the context of VQ-VAE usually means Exponential Moving Average, but the authors say it is Expectation-Maximization Attention. I cannot find this term in the original or other VQVAE paper, please clarify this point.

**Questions:**

- Is the citation for WavTokenizer generated by LLM?
- Line 123: To my knowledge, "EMA" in the context of VQ-VAE usually means Exponential Moving Average, but the authors say it is Expectation-Maximization Attention. I cannot find this term in the original or other VQVAE paper, could you clarify it?

---

> ### Author Response · Authors · 2025-11-20
>
> Thank you very much for your review. We believe we have addressed all your concerns and hope that you will consider raising your rating.
> 1. Experimental concerns regarding FSQ and Q2D2 comparison
>  We have addressed this by adding a direct, controlled comparison between Q2D2, FSQ, and VQ, all implemented within the same WavTokenizer framework. This ensures fairness and isolates differences strictly to the quantization layer - please see table 4 in the new uploaded version of the paper.
>
> 2. Missing baselines
>  Following the reviewer’s suggestions, we expanded our evaluation to include comparisons against multiple state-of-the-art neural codecs, including X-Codec, X-Codec 2, StableCodec, Mimi, SemanticCodec, and others. All comparisons are now reported in the revised paper (table 3).
>
> 3. Improper usage of UTMOS
>  Thank you for the observation. We agree that UTMOS, being trained on clean speech, may yield higher scores for models that reduce background noise—particularly on noisy subsets such as LibriTTS test-other. Our intention was not to treat this as an advantage of Q2D2, but to report the metric consistently across all baselines as the same in WavTokenizer paper and similar papers.
>
> 4. Question 1 – Citation for WavTokenizer generated by LLM
>  Yes, the original citation was generated automatically by an LLM, as disclosed in the paper. We have corrected and replaced it with the proper citation in the revised version.
>
> 5. Question 2 – Clarification about “EMA” in Line 123
>  You are absolutely correct that in the VQ-VAE literature, EMA stands for Exponential Moving Average, the standard codebook update mechanism used in VQ-VAE and VQ-VAE-2. The term “Expectation–Maximization Attention” was incorrect and has been removed. Our confusion was because both Exponential Moving Average (EMA) and Expectation–Maximization (EM) procedures are referenced in VQ-VAE–related works as EM and EMA. The revised paper now accurately reflects the proper meaning and usage of EMA. Thanks for catching it.

---

### Note · Program_Chairs · 2026-01-17
**Submission Desk Rejected by Program Chairs**

The following references in this submission do not refer to real documents and/or have major errors in bibliographic information:

 Sunghwan Ahn, Beom Jun Woo, Min Hyun Han, and Nam Soo Kim. Apcodec: Advanced perceptual audio codec with convnextv2. arXiv preprint arXiv:2407.12345, 2024a.
Alaaeldin El-Nouby, Hugo Touvron, Mathilde Caron, Piotr Bojanowski, Armand Joulin, Matthijs Douze, and Hervé Jegou. Product quantization for transformers. In International Conference on Machine Learning (ICML), 2022. URL https://arxiv.org/abs/2209.14509.
Hubert Siuzdak, Paweł Drozdowski, Christian Rathgeb, and Christoph Busch. Voice activity detection and classification using convolutional neural networks. IEEE Access, 6:2441-2450, 2018. doi: 10.1109/ACCESS.2017.2786642.